# How Has the COVID-19 Pandemic Changed Urban Consumers’ Ways of Buying Agricultural Products? Evidence from Shanghai, China

**DOI:** 10.3390/healthcare10112264

**Published:** 2022-11-11

**Authors:** Zengjin Liu, Jing Zhao, Zhuo Yu, Zhou Zhou, Liyuan Wang, Yusheng Chen

**Affiliations:** 1Shanghai Academy of Agricultural Sciences, Shanghai 201403, China; 2School of Management, Ocean University of China, Qingdao 266100, China

**Keywords:** COVID-19 pandemic, consumer behavior, consumption channel of agricultural products, influence factors

## Abstract

The COVID-19 pandemic has had a huge impact on people’s consumption behavior and habits. This paper takes Shanghai, China as a case study and uses a questionnaire survey of urban residents in all districts in Shanghai from April to May in 2022. Herein, we empirically analyze the factors affecting shopping modes for agricultural products, describe how things have changed compared to before the outbreak of the pandemic, and explore the underlying mechanism. This paper can provide a policy reference for how to ensure the safety of people’s food supply in the context of the COVID-19 pandemic. The results show that urban residents pay more attention to the basic attributes of agricultural products such as the quantity guarantee and health safety, and will adjust their consumption modes for agricultural products according to the needs of families and management. Compared with shopping malls and supermarkets, the quantity and quality assurance of community groups and e-commerce platforms can better meet the consumption situation of agricultural products during the pandemic period. The moderating effect of consumer commodity preference in the positive influence of safety evaluation on the transformation of shopping mode is significant. In general, online e-commerce platforms and community group buying have played a significant role in ensuring the acquisition of supplies needed by people during the COVID-19 pandemic. However, it remains to be determined whether the changes in the shopping modes for agricultural products of urban consumers brought by the current COVID-19 pandemic are long term or solidified.

## 1. Introduction

Agricultural products can provide human beings with the materials and energy necessary for life. Usually, people will access the agricultural products they need in supermarkets, farmer’s markets, and other places. With the development of the internet, people can also access their needed food through online e-commerce platforms (such as Fresh Hema, Ding Dong, etc.) and community group buying (community support for agriculture, cooperative distribution, etc.) easily and quickly. With the increase of various shopping modes, consumers choose a certain way of buying agricultural products according to their own experience and preferences. Under normal circumstances, consumers will choose a certain way of purchase based on the objective factors of shopping convenience and weather [1,2]. However, the outbreak of the COVID-19 pandemic in 2020 has had a huge impact on people’s lives around the world, especially people’s consumption modes for agricultural products [3]. In order to ensure people’s health and safety, the Chinese government has implemented strict prevention and control measures in the affected areas, which have achieved remarkable results. Globally, the impact of the COVID-19 pandemic has had a significant impact on consumers’ agricultural purchases and has changed the way that Chinese consumers, especially urban consumers, buy products. In recent years, the rapid development of China’s online e-commerce platforms has largely alleviated the problems faced by urban consumers buying agricultural products during the COVID-19 pandemic, bringing them many conveniences. During the COVID-19 pandemic, urban consumers have faced a lot of inconvenience in purchasing agricultural products, and the safety evaluation of agricultural products has also changed. Online e-commerce platforms and community group buying can realize “online platform booking, payment for goods, pick up goods at designated places, and realize the whole process of ‘no contact’”, which plays a positive role in assisting citizens to ensure the provision of living materials [4,5]. Therefore, exploring the influencing factors of urban consumers’ choice of consumption channel for agricultural products under the impact of COVID-19 and changes in purchasing channels compared to before COVID-19 is of great importance for ensuring the supply of agricultural products during the pandemic and further stabilizing the supply and market of agricultural products in the post-pandemic era.

Different shopping modes can bring consumers different shopping experiences. The consumer preferences exposed by consumers under their limited choices reflect the reasons why they choose a certain way of shopping, and also reflect the advantages and disadvantages of various kinds of shopping modes to guarantee supply in special periods. In the process of supermarket shopping, consumers can measure the quality of fresh agricultural products by personally touching the product and observing the color. Online shopping platforms can accurately and quickly push relevant products to consumers according to people’s consumption habits and preferences [6]. By constantly improving the shopping system and recommending relevant products according to customers’ preferences, online e-commerce platforms enable consumers to obtain the products they need more directly and efficiently [7,8]. In addition, the delivery service of online shopping platforms saves shopping time, and the quality of their delivery service also significantly affects consumers’ purchase intentions [9]. However, consumers of online shopping can only obtain product information through the limited pictures and text displayed on the online platform, which leads to greater perceived risks [10]. One of the main reasons why some consumers reject online shopping is the fear that there is a big gap between the actual quality and safety perception of products and their high expectations [11,12]. Community group buying relies on the mode of “e-commerce + social networking”. The leader establishes the community wechat groups and regularly releases group buying information, gathering the consumers in the community with the same needs. In this way, people buy the products online with a large discount through the applications [13]. The leader, a position generally held by people with high credibility in the community, helps to solve the problem of unqualified agricultural products in the process of online sales [14].

The research shows that consumers’ personalities, family characteristics, shopping experience, purchasing motivation, attributes of purchased products, and services offered by purchasing platforms may all affect people’s choice of purchasing method [15,16,17,18,19,20]. Consumers’ willingness to buy agricultural products online is influenced by gender and education level [21]. Consumers’ age, income, and whether they are married also affect their willingness to buy agricultural products online [17,22]. For consumers who buy fresh agricultural products, the product quality and logistics quality of e-commerce platforms significantly affect consumers’ willingness to buy [23,24]. Consumers’ familiarity with websites also affects their online purchase intentions [25,26]. Furthermore, with people’s pursuit of healthy food, whether the product itself is a healthy food also affects people’s shopping decisions [27]. Consumers pay great attention to the quality of agricultural products in the process of purchasing. Green agricultural products with third-party certification labels are more likely to be selected by consumers [28,29,30,31]. In the context of the COVID-19 pandemic, people’s food consumption behavior has changed [32,33,34,35,36,37]. During the pandemic, the frequency of offline food purchase has decreased [38], and the frequencies of online shopping, non-contact distribution services, and other new consumer behaviors have increased [39]. Most previous studies have focused on one or two types of consumption, therefore failing to detect changes in all consumption behaviors for urban consumers during the COVID-19 pandemic. In particular, no research has been carried out on the changes and driving factors of fresh agricultural product consumption patterns of urban consumers during the pandemic. The analysis of these changes and driving factors can provide policy ideas for ensuring the supply of fresh agricultural products for people living in big cities during the pandemic. In addition, in the context of the pandemic, understanding the consumption habits of urban consumers will help suppliers of fresh agricultural products adjust the supply chain of fresh agricultural products. In March 2022, the COVID-19 pandemic broke out in Shanghai on a large scale. In order to prevent and control the pandemic, Shanghai has entered the state of global static management. In this context, it is of great significance and value to study the changes and causes of agricultural product shopping modes of urban consumers in Shanghai, the international metropolis. Accordingly, taking Shanghai, China as a case study and using a questionnaire survey of urban consumers in all districts in Shanghai during the pandemic, the changes in agricultural product shopping modes of urban consumers under the impact of COVID-19 are revealed. In this paper, we empirically analyze the main factors affecting consumers’ choice of agricultural product shopping modes, discuss the factors that can cause consumers to change their purchasing channel compared to before the outbreak of the pandemic, and explore the underlying mechanism. The contribution of this study is to discover and explain the changes in the way urban consumers buy agricultural products brought about by the COVID-19 pandemic, and to explain the reasons mainly from the perspective of agricultural product safety evaluation. This can not only provide a policy reference for how to ensure the safety of people’s food supply in the context of the COVID-19 pandemic, but also provide Shanghai’s “solution” for other regions to emulate. The content of this study is of great value to food suppliers, and can provide information for them to adjust their sales methods in a timely manner in the context of the pandemic. It remains to be determined whether the changes in the shopping modes of agricultural products of urban consumers brought by the current COVID-19 pandemic are long term or solidified.

## 2. Theoretical Analysis and Model Building

### 2.1. Theoretical Analysis

There are many factors that affect consumers’ choice of shopping modes. During the COVID-19 pandemic, the decision-making process of consumer purchasing methods is not only influenced by their personal characteristics such as gender, age, and education [40], but also influenced by the types of agricultural products available for selection, the quality of agricultural products, and the distribution time. In an emergency, consumers will behave differently. In order to avoid food shortage or price increases in the future, many consumers will hoard a large number of products through various shopping modes to meet their daily needs [41]. Often, online promotional offers play a key role in influencing consumer buying behavior [42,43]. Changes in household income and agricultural product prices during the pandemic period may also affect consumers’ choice of purchasing channels. In addition, the duration of community static management and the timeliness of help provided by the government also affect consumers’ choice of shopping modes.

In the 1960s, American scholar E. S. Lee proposed the migration theory, which divides the factors affecting population migration into two types: push factor and pull factor [44]. The push factor is due to the higher cost of living and taxes in the place of relocation, which is a negative factor. The pull factor is caused by lower living costs and taxes in places where people move, which is a positive factor. In addition, during the process of population migration, people have a herd mentality because of the decisions of the surrounding population, and they will also maintain the status quo because of the cost of migration. These factors are known as the “anchor force”. This theory has been widely applied since being proposed, especially in the study of consumer behavior [45]. Comparing before and after the COVID-19 pandemic, changes in agricultural product shopping modes of urban consumers in Shanghai constitute consumer migration behaviors.

The factors that affect the transformation of consumers’ purchasing pattern involve convenience, price et.al. [46,47]. Some scholars point out that although consumers can easily buy goods through online channels, offline stores such as supermarkets are still consumers’ first choice [48]. However, during the COVID-19 pandemic, when there is a shortage of supplies, there exist push factors in their decision whether to change their shopping modes because of concerns about inadequate food at home and limited options for food [49]. In other words, “safety evaluation” (ensuring the supply of household goods) has been the push force in changing people’s shopping modes. Consumption preference refers to the subjective feeling and judgment of the products or services purchased by individuals, which influences the purchase intention of consumers [50,51]. The research of Cang et al. showed that different types of consumers are very concerned about the quality of fresh agricultural products, the quality of logistics services, reputation, and the accuracy of website information [52]. Especially during the COVID-19 pandemic, due to the inconvenience of community management and control as well as safety concerns, more and more people are unwilling to venture out to buy necessities. As a result, consumers’ shopping modes for agricultural products have changed to online shopping with logistics guarantees [3]. The “consumption preferences” of people during the COVID-19 pandemic have been the pull factors for the shift in their shopping modes. In addition, this shift also requires a certain “cost” [53], i.e., the “anchoring force” in the process of changing shopping modes. Age and educational qualifications affect online shopping [22]. Residents who are unfamiliar with network operation and online shopping platforms need to spend a lot of time learning related operations [54]. Therefore, elements such as age and educational background can influence the shift in people’s shopping modes.

According to the above analysis, combining consumer preference theory, migration theory, risk perception theory, and other research in related fields, this paper takes into consideration the factors that affect the choice of agricultural product shopping modes of Shanghai urban consumers during the COVID-19 pandemic and the change of shopping modes compared to before the pandemic. The theoretical model framework has been constructed as shown in Figure 1.

### 2.2. Constructing the Model

#### 2.2.1. MProbit Model

The traditional Probit Regression Model is usually aimed at the dichotomous dependent variable, whereas the Multi-classification Probit Regression Model is suitable for the situation where the dependent variable has more than two values. The Multi-classification Probit Regression Model can be classified into two types: Multinomial Probit Regression Model (MPR) and Ordinal Probit Regression Model (OPR). Although there is no logical order of the agricultural product shopping modes chosen by consumers, there may be correlations in the disturbance terms. Given these characteristics of the dependent variable in this paper, MPR is applied for joint estimation.
(1)$$\begin{array}{l} Y_{j}^{1}{}^{*}=\beta_{1,i}*x_{i}+\varepsilon_{1}\\ Y_{j}^{2}{}^{*}=\beta_{2,i}*x_{i}+\varepsilon_{2}\;\\ Y_{j}^{3*}=\beta_{3,i}*x_{i}+\varepsilon_{3} \end{array}\;,\;Y=\left\{\begin{array}{l} 1\;\;\;\;\;\;\;if\;\;\;\;Y_{j}^{1*}>Y_{j}^{2*}\;,\;Y_{j}^{3*}\\ 2\;\;\;\;\;\;\;if\;\;\;\;Y_{j}^{2*}>Y_{j}^{1*}\;,\;Y_{j}^{3*}\\ 3\;\;\;\;\;\;\;if\;\;\;\;Y_{j}^{3*}>Y_{j}^{1*}\;,\;Y_{j}^{2*} \end{array}\right.\;\;$$

Specifically, during the COVID-19 pandemic, customers’ shopping modes mainly include “supermarkets”, “e-commerce platforms”, and “community group purchasing” (m = 1, 2, 3). $Y_{j}^{1*}$, $Y_{j}^{2*}$, and $Y_{j}^{3*}$ in Formula (1) correspond to the latent variables of the above three behaviors, respectively; j can take the value of 1, 2, or 3, corresponding to the chosen control group. ε is the corresponding residual term. At this time, the basic equation under the joint estimation principle of this model is as follows:(2)$$f_{m}=\ln\left(\frac{H(Y=m)}{H(Y=j)}\right)=x_{i}{}^{\prime }*\beta_{{}_{m,i}}$$

In accordance with the interest and focus of this paper, “supermarkets” is chosen as the control group (j = 1), with pairwise coupling. Three bivariate Probit models are constructed; in other words, we can obtain the joint estimation system. In Formula (3), $\beta_{1,1}$ is equal to 0; $\beta_{2,1}$ and $\beta_{3,1}$ are parameters to be estimated; $x_{i}$ (i = 1, 2, …, 11) represents factors that affect agricultural product shopping modes and control variables. Y (1, 2, 3) represents the main purchase methods for agricultural products by consumers during the COVID-19 outbreak. The result is assumed to follow the multivariate normal distribution, and H (Y) is the expression under the corresponding Y value. The predicted probability of each choice can be obtained by solving three formulas:(3)$$\left\{\begin{array}{l} f_{1}=\ln\left(\frac{H(Y=1)}{H(Y=1)}\right)=\ln1=x_{i}{}^{\prime }*\beta_{{}_{1,1}}=0\\ f_{2}=\ln\left(\frac{H(Y=2)}{H(Y=1)}\right)=x_{i}{}^{\prime }*\beta_{{}_{2,1}}\\ f_{3}=\ln\left(\frac{H(Y=3)}{H(Y=1)}\right)=x_{i}{}^{\prime }*\beta_{{}_{3,1}} \end{array}\right.$$

#### 2.2.2. Probit Model

In order to explore the factors affecting the change in specific shopping modes, it is assumed that whether the change occurs or not is determined by the potential utility ($U_{*}$). When the utility (U) is lower than or equal to ($U_{*}$), consumers will not make such a change in shopping mode; otherwise, such a change will occur.
(4)$$\begin{array}{l} \mathrm{Pr}obit(Y_{n}=1)=\mathrm{Pr}obit(U_{n}>U_{{}_{n}}^{*})\\ \mathrm{Pr}obit(Y_{n}=0)=\mathrm{Pr}obit(U_{n}\leq U_{{}_{n}}^{*}) \end{array}$$

The latent variable, the utility, is determined by factors such as safety evaluation:(5)$$U_{n}=\beta_{0}+X_{i}\beta_{n,i}+\mu_{n}$$

Assuming that the multivariate normal distribution applies, the corresponding probability density function is
(6)$$\mathrm{Pr}obit(Y_{n})=\phi (\beta_{0}+X_{i}\beta_{n,i})$$

With each shift in the shopping modes, the Probit Mode is
(7)$$Y_{1}=f(x_{i},\mu_{1})\;\;,\;\;Y_{2}=f(x_{i},\mu_{2})\;\;,\;\;Y_{3}=f(x_{i},\mu_{3})\;\;$$
where $Y_{n}$ (*n* = 1, 2, 3) corresponds to “from supermarkets to e-commerce platforms”, “from supermarkets to community group purchasing”, and “from e-commerce platforms to community group purchasing”, respectively.

The definitions of variables and basic statistical characteristics are shown in Table 1.

## 3. Data Source and Descriptive Statistics Analysis

### 3.1. Data Source and Sample Features

Data in this paper are from our group’s online survey on Shanghai urban consumers’ consumption of agricultural products during the period of pandemic prevention and control in April and May, 2022. Aided by the mini program Wenjuanxing, the questionnaire survey was carried out with a total of 1055 questionnaires collected. After the process of logic check, 1046 valid questionnaires were obtained, with effective questionnaires accounting for 99.15%. The characteristics of the samples are shown in Table 2. In terms of gender, the proportion of male versus female respondents is relatively balanced, with each accounting for around 50%. In terms of age, the respondents were mainly young and middle aged, with young people aged 20–39 and middle-aged people aged 40–59 accounting for 54.22% and 41.42%, respectively. To some extent, this shows that the sample can effectively match with the overall population in terms of gender, age characteristics, income changes, etc. Again, this leads us to some possible speculations. In terms of family division of labor, there is no significant gender difference in purchasing agricultural products among urban consumers who are mainly middle aged and young. In addition, during the COVID-19 pandemic, the income level of many consumers has declined. Consumers may have to choose to use some of their savings or reorganize portfolio to cope with unexpected external challenges. Under the influence of a superposition of supply conditions, the overall consumption level of agricultural products may decline to some extent. In terms of educational qualification, the respondents were generally well educated, among which 72.13% respondents had acquired a bachelor’s degree or above. To some extent, this fact reflects the ability of Shanghai as a megacity to attract high-quality talents.

### 3.2. Features of Urban Consumption of Agricultural Products and Changes in Shopping Modes under the Impact of the COVID-19 Pandemic

Six features can be obtained according to the analysis of questionnaire data. It is clear that the COVID-19 pandemic has had a certain impact on the supply market of agricultural products. In general, the prices of agricultural products have risen. Agricultural products whose prices have risen by 10% and 49% account for 65.68%. Secondly, due to e-commerce platforms under obvious pressure, more than 50% of the respondents said that they often encounter problems such as “difficult to order vegetables”. Thirdly, the phenomenon of tight or insufficient supply of agricultural products is prominent in vegetable and fruit supply, with a ratio reaching 80.95%. Furthermore, due to the impact of necessary measures such as static management, the shopping modes for agricultural products have changed significantly in the short term (compared to before the pandemic), resulting in “consumption downgrade”. First of all, the proportion of offline shopping modes has decreased rapidly (Figure 2); the proportion of agricultural products mainly purchased through offline channels has decreased from 68.63% before the pandemic to 16.40%. Secondly, the proportion of community group purchasing and e-commerce platforms has increased significantly (Figure 2), with the proportion of community group purchasing increasing from 14.41% to 59.24% and the proportion of e-commerce platforms increasing from 16.78% to 23.51%. Thirdly, consumers’ preferences for commodities have changed from high-level needs (taste, freshness, etc.) before the pandemic to basic attributes, such as quantity assurance, guaranteed delivery, and hygiene safety under realistic constraints.

## 4. Model Estimation Results and Analysis

In this part, we mainly elaborate on the purchase channels of agricultural products and changes therein, illuminating how safety evaluation and preference heterogeneity have affected the way of buying agricultural products during the COVID-19 pandemic. According to the model selection, variable setting, descriptive statistics, and other results mentioned above, Stata16 software was adopted to estimate the multivariate disordered Probit and the binary Probit model involved in the following two parts, with the Maximum Likelihood principle.

### 4.1. The Influence of Safety Evaluation and Other Factors on the Shopping Modes of Agricultural Products during the Pandemic

It can be concluded from Table 3 that, overall, the model is highly significant given the wald test statistic and *p*-value. Under the premise of taking the supermarket as the benchmark group and considering the control variables such as the timely support of the government and community institutions, for the explanatory variables, some divergences exist in the significance of two different shopping modes: e-commerce platforms and community group purchasing. From the perspective of how variables work, compared with the mode of going shopping at the supermarket, consumer groups who buy agricultural products online are dramatically impacted by the noteworthy positive influence of quantitative safety evaluation, household income changes, and the duration of static management, whilst being dramatically impacted by the significant negative influence of the preference of commodity quality. There are still problems for e-commerce platforms in meeting consumers’ requirements for the agricultural product quality. In comparison to purchasing products in the supermarket, consumers in the city choose to obtain agricultural products through community group purchasing, which is primarily influenced by the significant positive effect of the quality safety evaluation, the price increase of agricultural products, the preference for quantity of commodity, and static management duration. As for effects of variables, only the average value of marginal effects of some variables have been analyzed and compared due to the particularity of the nonlinear regression equation in contrast with the linear regression equation. The significant variables above are relatively similar in the degree of their effects without remarkable disparity, and their direction of effects is relatively consistent with the previous estimation results.

Therefore, to some extent, we can infer that by comparison with the supermarket, community group purchasing and shopping through e-commerce platforms have demonstrated competitive advantages in the quantity and price of agricultural products during the pandemic period. Moreover, in order to make the research results of the above selective behaviors of purchases more explicit, it is essential to carry out further empirical research on the intra-group influence mechanisms of the representative (several types accounting for a large proportion of the sample) transformations of shopping modes from the perspective of safety evaluation and preference.

### 4.2. Analysis of the Influential Mechanisms of Safety Evaluation and Other Factors in the Transformation of Shopping Modes during the Pandemic

The following conclusions can be clearly drawn from the estimation results in Table 4. In the process that security evaluation affects the transformation of specific shopping modes, commodity preference plays a significant role in moderating. Other factors also have some heterogeneity in this progress. When consumers make purchases from the e-commerce platforms rather than from supermarkets, interactive items between the preference for commodity quantity and quantity safety evaluation have a significant positive effect, which indicates that, in this specific process, consumers switch to e-commerce platforms from supermarkets, primarily due to the shortage of agricultural products caused by the pandemic. In other words, with this method, the strategy of storing enough food can be achieved more easily in special periods. Simultaneously, unstable family income during the pandemic also greatly impacts this conversion. Under the pandemic, the economy has slowed down somewhat, causing temporary problems in employment, management, and other aspects, thereby aggravating the fluctuation of household income of citizens; indeed, most family incomes show a downward trend. Hence, the consumption capacity is different provisionally, and then the demand for agricultural products of some consumers is changed. Compared with supermarkets, e-commerce platforms provide more diversified choices and price combinations of agricultural products.

When people purchase products through community group purchasing instead of supermarkets, the positive moderating effect of the preference of commodity quantity to quantity safety evaluation remains significant. The more household income increases, the more commodity quality is preferred. This means that if worrying about product quality, some residents are more likely to maintain their original consumption habit of buying agricultural products through supermarkets rather than through community group purchasing. Furthermore, the duration of static management has a significant positive effect on this shift. Citizens are more likely to buy products through community group purchasing if the static management lasts longer. Combined with the analysis of regional agricultural supply levels and management requirements, this situation may be related to the phenomenon that the short-term demand for agricultural products has spiked during the COVID-19 pandemic; the market cannot meet all consumption needs, and the relatively limited capacity of e-commerce platforms has something to do with it.

To explain this conclusion intuitively and compare the motivation of citizens comprehensively, the influential mechanisms of the shopping modes (e-commerce platforms transforming into community group purchasing) are further expounded here. First of all, quality safety evaluation, agricultural product price growth, household income changes, and static management duration have significant positive effects on this transformation. Consequently, community group purchasing can better meet the needs of consumers in the city for quality and safety, which is in accordance with the function of community group purchasing: to alleviate information asymmetry. Group purchasing based on community workers and community relations is a relatively reliable method for consumers to buy agricultural products, which raises the immoral cost of organizers as well as the reputation incentive. In this sense, supervision over the quality of agricultural products is strengthened. Meanwhile, requirements of group purchasing for consumers are more loose and friendly than that of shopping on the internet.

With the impact of the pandemic, the short-term supply of agricultural products is tight and their prices are rising. Confronting this phenomenon, some consumers in the city are more inclined to depend on community group purchasing for the limited price increase gap, to balance their needs for the quality of agricultural products with changes in family income. In the meantime, when the duration of static management grows longer, more opportunities to work together and communicate with each other in the community also make it possible to build stronger ties in the community. When the general demands of groups cannot be fully covered by the limited market supply, the willingness and behavior of community members to form a group-assisted organization to purchase products will grow. Moreover, with the intention to improve service during static management, community workers will also coordinate and organize individuals in the community to establish a purchase group for public welfare. This is why citizens are more willing to purchase directly through the community when the duration of static management increases. In addition, the moderating effect of preference for agricultural products safety evaluation has notable differences. Distinct from the quantity preference mentioned above, in this kind of transformation, quality attribute preference plays a significant and slightly negative role in the positive impact of quality safety evaluation on this change of purchase mode.

## 5. Discussion

From the perspective of shopping modes for agricultural products and transformations therein, this paper has conducted theoretical analysis and empirical verification of the influence of safety evaluation, preference heterogeneity, and other factors on urban consumers’ consumption of agricultural products during the pandemic, with 1046 questionnaires collected during the outbreak of the COVID-19 pandemic in Shanghai in 2022. The research findings can be summarized as follows: First, the consumption of agricultural products is affected to some extent by the pandemic. It was also evidenced by Butu et al. [55] that the consumption of urban consumers has been “degraded”, and attention has shifted to the basic requirements of agricultural products such as quantity assurance, health, and safety, and the consumption modes for agricultural products will be adjusted in accordance with the needs of families and management. This is in line with the findings of Eger’s research that the COVID-19 pandemic has changed customers’ behaviors and influenced traditional and online shopping [56]. Second, in terms of purchasing agricultural products during the pandemic period, community group purchasing and e-commerce platforms have shown more competitive advantages in quantity and quality compared to shopping at supermarkets, allowing these modes to better keep up with the demand for agricultural products and cater to the preferences of consumers. Similar results were found by Ellison et al. and Chenet al. [57,58]. During the period of COVID-19, people’s offline purchasing has decreased, while online purchasing has increased. However, there are some differences in the factors that play a significant role in the process of being selected by consumers. Third, for changes in shopping modes for agricultural products during the pandemic, with the analysis of three main shopping modes, the moderating effect of consumers’ commodity attribute preference on the positive impact of safety evaluation on the change of purchase mode is significant. The reason for the transition of shopping through community group purchasing from shopping through e-commerce platforms is that community group purchasing can satisfy consumers’ preference for commodity quality better under the significant influence of static management duration, family income changes, and agricultural product price increases. This mode can fit the common demand of community groups effectively, in the form of mutual assistance or volunteering, and it also mitigates the impact of the pandemic on household agricultural product consumption. Based on the above research conclusions, it is not difficult to see that major public health safety events such as COVID-19 have a certain impact on the purchase of agricultural products. Changes in the main purchase mode of urban consumers for agricultural products not only represent the passive adaptation of citizens under safety and health management regulations and other measures, but also citizens’ active response after changes in their own income and other conditions. From the above conclusions, the role of direct supply through group purchasing and e-commerce platforms in this respect cannot be ignored. Similarly, for agricultural producers and distributors, the impact of the changes in consumer preferences and safety assessment mentioned in the above conclusions should be taken into account in how to defuse risks and turn crises into opportunities. Of course, this research has some limitations; we still need to think about some problems. Is the shift in the way consumers buy agricultural products sustained or short lived due events such as COVID-19? If it is long term, on the one hand, we must consider whether the change improves the ability of the system to respond effectively to such shocks. On the other hand, we must ask how this will affect the producers and operators of agricultural products, that is, questioning how the production side can effectively respond to this change to stabilize or improve their revenue. These questions may require more in-depth and continuous follow-up studies.

## 6. Conclusions

This paper has conducted an empirical analysis based on a questionnaire survey of consumers in all urban areas of Shanghai during the COVID-19 pandemic from April to May in 2022. The COVID-19 pandemic has dramatically impacted consumers’ consumption behaviors. Urban consumers now pay more attention to the basic attributes of agricultural products such as quantity guarantee and health security, and will adjust the consumption mode for agricultural products according to family and management needs. Compared with the supermarket purchase mode, community group purchasing and e-commerce platforms can better satisfy the agricultural product consumption situation during the pandemic period in terms of quantity and quality assurance. The moderating effect of consumer commodity preference in the positive influence of safety evaluation on the transformation of shopping mode is significant. With reference to the above research conclusions, the following insights can be obtained: First, when dealing with the insufficient supply of agricultural products and the relative stoppage of traditional sales markets caused by the pandemic, government departments should organize or introduce more effective sales channels such as e-commerce platforms and community group purchasing to guarantee stable supply. Second, it is imperative to meet citizens’ preference for quantity of agricultural products as much as possible. Meanwhile, as a prerequisite to relieve the public panic, the preference for quality should be satisfied to a certain extent, with the purpose of achieving static management objectives. Third, in the context of the pandemic, the main producers of agricultural products should adopt more proactive strategies to address market risks and try non-traditional marketing channels and models to establish a more stable relationship of production and marketing. Fourth, business entities of agricultural products should improve their service and adopt appropriate publicity strategies, so as to strengthen consumer trust and dependence and also to maintain the sustainable transformation of purchase channels.

## Figures and Tables

**Figure 1 healthcare-10-02264-f001:**
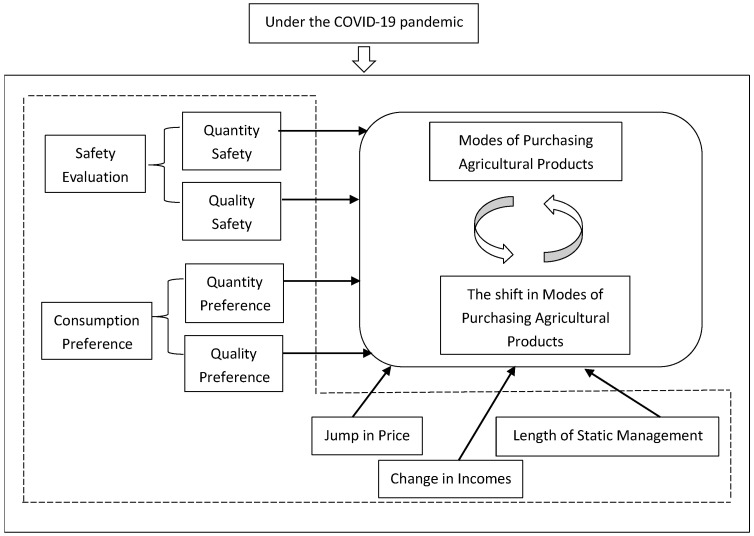
The theoretical model for changes in urban consumers’ ways of buying agricultural products.

**Figure 2 healthcare-10-02264-f002:**
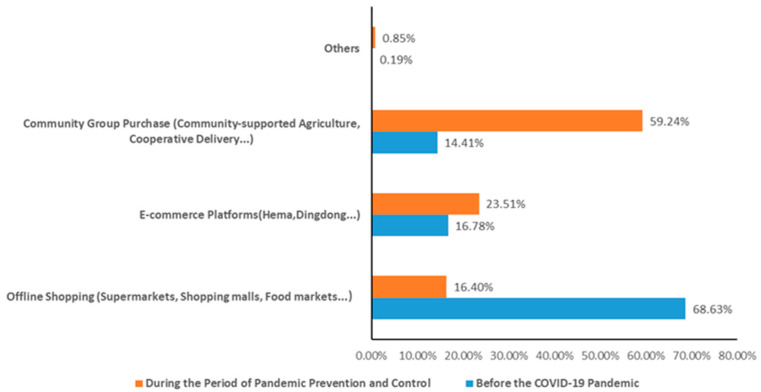
Changes in the purchase modes for the main agricultural products of households.

**Table 1 healthcare-10-02264-t001:** Setting and meaning of variables in MProbit and Probit Model.

Variables	Index	Assignment Instruction	Average	Standard Deviation
Explained Variables				
Modes of Purchasing Agricultural Products	The main mode of purchasing agricultural products during the COVID-19 pandemic	The main mode: supermarkets = 1, e-commerce platforms = 2, community group purchasing = 3	2.432	0.759
The Shift in Modes of Purchasing Agricultural Products	Does it change from “supermarkets” to “e-commerce platforms”?	Yes = 1, No = 0	0.154	0.361
	Does it change from “supermarkets” to “community group purchasing”?	Yes = 1, No = 0	0.401	0.490
	Does it change from “e-commerce platforms” to “community group purchasing”?	Yes = 1, No = 0	0.079	0.270
Explanatory Variable				
Safety Evaluation	Quantity Safety	During the static management period, do you agree or disagree with the statement such as worrying about the lack of food at home? Strongly Disagree = 1, Disagree = 2, Somewhat Disagree = 3, Neutral = 4, Somewhat Agree = 5, Agree = 6, Strongly Agree = 7	5.427	1.424
	Quality Safety	Did your family have a limited choice for food varieties during the period of static management? Strongly Disagree = 1, Disagree = 2, Somewhat Disagree = 3, Neutral = 4, Somewhat Agree = 5, Agree = 6, Strongly Agree = 7	5.141	1.642
Jump in Price	Jump in the Price of Agricultural Products	How much higher than usual are the prices of agricultural products you buy? Lower than 10% = 1, 10–19% = 2, 20–29% = 3, 30–49% = 4, 50–99% = 5, More than 100% = 6	3.211	1.508
Change in Income	Change in Family Income	During the period of static management, how has your family income been? Much lower = 1, somewhat lower = 2, normal = 3, somewhat higher = 4, much higher = 5	1.868	0.744
Length of Static Management	Duration of Static Management	0 day = 1, 1–10 day(s) = 2, 10–20 days = 3, 20–30 days = 4, 30–40 days = 5, more than 40 days = 6	3.778	1.491
Consumption Preference	Quantity Preference	During the period of static management, have you considered the elements of “fast delivery” as well as “products guaranteed” when you buy agricultural products? Yes = 1, No = 0	0.563	0.496
	Quality Preference	During the period of static management, have you considered “hygiene safety” when you buy agricultural products? Yes = 1, No = 0	0.781	0.414
Control Variables				
Gender		Your gender: Male = 1, Female = 2	1.503	0.500
Age	Your age	Actual Age (Integer)	36.162	9.120
Educational Background	Educational Qualification	Your educational background: Primary School = 1, Junior Middle School = 2, Senior High School (Secondary Technical School) = 3, Undergraduate College = 4, Master’s Degree or Above = 5	4.675	0.820
Governmental Security	Timeliness of Assistance from Government and Communities	When you have difficulty in daily life, you can get help from your community. Strongly Disagree = 1, Disagree = 2, Somewhat Disagree = 3, Neutral = 4, Somewhat Agree = 5, Agree = 6, Strongly Agree = 7	4.875	1.435

**Table 2 healthcare-10-02264-t002:** Basic characteristics of Shanghai agricultural product consumers.

Item Name	Option	Proportion
Gender	Male	49.86%
	Female	50.14%
Age	<20	3.70%
	20–39	54.22%
	40–59	41.42%
	≥60	0.66%
Educational background	Primary School	0.28%
	Junior Middle School	1.80%
	Senior High School (Secondary Technical School)	7.77%
	Undergraduate College	64.55%
	Master’s Degree or Above	7.58%
Change in Income	Much Lower	33.27%
	Lower	48.06%
	Normal	17.44%
	Higher	0.95%
	Much Higher	0.28%

**Table 3 healthcare-10-02264-t003:** MProbit model regression results (with supermarket as the benchmark group).

Variables	E-Commerce Platforms = 2	Community Group Purchasing = 3
Safety Evaluation		
Quantity Safety	0.111 *	0.056
	(0.058)	(0.054)
Quality Safety	0.059	0.100 **
	(0.051)	(0.048)
Consumption Preference		
Quantity Preference	0.125	0.302 **
	(0.150)	(0.141)
Quality Preference	−0.437 **	−0.169
	(0.182)	(0.176)
Jump in Price	−0.002	0.099 *
	(0.061)	(0.057)
Change in Income	0.174*	−0.159
	(0.101)	(0.097)
Length of Static Management	0.229 ***	0.360 ***
	(0.053)	(0.050)
Age	0.012	0.032 ***
	(0.008)	(0.008)
Gender	0.295 *	0.580 ***
	(0.150)	(0.141)
Educational Background	0.111	0.240 ***
	(0.089)	(0.085)
Timeliness of Assistance from Government and Communities	0.074	0.086
	(0.057)	(0.054)
Wald chi2(22) = 174.170		
Prob > chi2 = 0.000		

Note: The numbers outside the brackets are estimated coefficients, and the figures in the brackets are standard errors; ***, **, and * are levels of significance for 1%, 5%, and 10%, respectively.

**Table 4 healthcare-10-02264-t004:** Probit model regression results for three specific transformations of shopping modes.

Variables	From Supermarkets to E-Commerce Platforms	From Supermarkets to Community Group Purchasing	From E-Commerce Platforms to Community Group Purchasing
Safety Evaluation			
Quantity Safety	0.058	−0.029	0.069
	(0.041)	(0.034)	(0.058)
Quality Safety	−0.029	0.027	0.119 **
	(0.039)	(0.033)	(0.051)
Jump in Price	−0.044	0.020	0.087 *
	(0.038)	(0.032)	(0.049)
Change in Income	0.178 ***	−0.157 ***	0.172 *
	(0.067)	(0.058)	(0.086)
Length of Static Management	−0.043	0.065 **	0.149 ***
	(0.035)	(0.030)	(0.050)
Age	−0.004	0.011 **	0.005
	(0.006)	(0.005)	(0.008)
Gender	−0.057	0.061	0.591 ***
	(0.096)	(0.081)	(0.130)
Educational Background	0.038	0.074	0.088
	(0.062)	(0.052)	(0.084)
Timeliness of Assistance from Government and Communities	0.038	-0.009	0.042
	(0.037)	(0.030)	(0.045)
Moderating Effect			
Quantity Safety × Quantity Preference	0.032 * (0.018)	0.049 *** (0.015)	−0.030 (0.022)
Quality Safety × Quality Preference	0.018 (0.023)	0.027 (0.018)	−0.053 ** (0.024)
LR chi2(11)	22.46 **	59.21 ***	63.49 ***
Pseudo R2	0.025	0.042	0.110

Note: The numbers outside the brackets are estimated coefficients, and the figures in the brackets are standard errors; ***, **, and * are levels of significance for 1%, 5%, and 10%, respectively.

## Data Availability

The data presented in this study are available on request from the corresponding author.

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
