# Peer review of "How Has the COVID-19 Pandemic Changed Urban Consumers’ Ways of Buying Agricultural Products? Evidence from Shanghai, China"

_healthcare, 2022, doi:10.3390/healthcare10112264_

Round 1

Reviewer 1 Report

The paper is interesting. The topic related to consumption during the pandemic are increasingly present in the literature. It is advisable to arrange small things inside the paper.

-It is recommended to reduce the title, example: Does the COVID-19 Pandemic Change Urban Residents' Ways of Buying Agricultural Products? Evidence from Shanghai, China

-Insert lines 105-108 in the abstract to insert the contribution of the paper in the literature.

-Line 214, is it a thesis? Better to write paper or work. Maybe the results were taken from a thesis and forgot to delete it.

- Adjust the conclusions paragraph. Insert a discussion paragraph (lines 385-398), further supplementing both paragraphs.

- Insert further references.

Reviewer 2 Report

This manuscript is interesting as it aimed to study Chinese consumers in Shanghai during the COVID 19 pandemic.  With Chinese government policy and management, it posted a special circumstance to study human attitude and behavior changes when compared with the "old normal" period.   The study design was retrospective survey of 1,000+ Shanghai residence.   The authors started the study with a sound theoretical model and provide data and results to support and explain the observed behavior and attitude through the use of self-administer questionnaire.  There are minor typing errors and writing styles that need to be changed as below:

Detailed comments:

Page 4  Figure 1 title

Figure 1 should have a better title.  Such as: The Theoretical Model of ...

Page 5 Table 1 title

The Table title should be more specific.  For example, Setting and Meaning of variables in ...  model. 

Page 5  Line 183

What is the symbols used in m=1, 2, 3...?  Are they comma (,)?

Page 5  Line 188

The same strange symbol appears again.

Page 7  Table 2 title

The table's title should be more specific and descriptive.  For example, Basic characteristics of Shanghai consumers ... .

Page 7  Line 217-225

The writing here contains the same information as that in Table 2.  Please rewrite and focus on something that interesting and need some effort when reading Table 2.  Otherwise, you are repeating the information.

Page 8   Line 244-245

This writing is strange.  It is difficult to read.  Suggest "... from 14.41% to 59.24% and from 16.78% to 23.51%, respectively. 

Page 8  Figure 2

Recommend to change graph type to Bar chart that you can put "Before..." and "During..."  side by side.  That will help you to communicate the changes better.

Page 8 Line 250

I think this section title is wrong.  It should be 4.

Page 8  Table 3 title

No need to have "." at the end for this title as it is not a complete sentence or followed by another sentence.  Should indicate that you used Supermarket as the benchmark group.

Also, the table should have lines separating the variable.  In current format, it is very difficult to read the coefficients and their corresponding standard errors for the variables.

Page 9  Line 263-264

This sentence does not make any sense.  What was "in line with the estimation...".  You can just say that the mode was highly significant (test statistic and p-value).  

Page 10  Table 4 title

 The table title need to be more descriptive.

The table should have lines to separate the variables and corresponding means and SEs.  Right now, it is very difficult to read.

Page 11  Line 298-300

The language used for the first sentence of this paragraph is strange

Page 11  Line 304

Need "," before "in". "...that, in this specific process,..."

Reviewer 3 Report

I begin by congratulating the authors and researchers of the manuscript intitled “Does the COVID-19 Pandemic Change Urban Residents' Ways of Buying Agricultural Products? --Based on Questionnaire Evidence from Shanghai, China During the Pandemic”. The topic is interesting.

Personally, I disagree with the actual title. As suggestion, something like “How the Covid-19 pandemic change the way of Shanghai consumers buy agricultural products” could be more obviously informative and clear.

Use the term “article” instead of “paper”, along all the manuscript.

Add the limitations of the study, and also the future research part too taking in account previous papers of Healthcare journal.

Good luck!

Reviewer 4 Report

Dear Authors,

This time, I will try to give my comments in a more clear version:

The author needs to clarify the new contribution of the research in the introduction. It is necessary to clearly state the new and motivating points of the article.

The literature review should be placed after the missing section. Authors need to update recent studies. And point out the missing point to carry out this study. The author should have a literature review to compare the results of previous studies conducted in the same research context.

The author needs to show the research equation clearly, the measurement variables.

The author needs to make a clear hypothesis for each pair of variables.

I hope my comments may help you in developing the paper.

Round 2

Reviewer 4 Report

The author has addressed all my comments.